# Development and Characterization of an Edible Zein/Shellac Composite Film Loaded with Curcumin

**DOI:** 10.3390/foods12081577

**Published:** 2023-04-07

**Authors:** Tao Han, Wenxue Chen, Qiuping Zhong, Weijun Chen, Yaping Xu, Jiawu Wu, Haiming Chen

**Affiliations:** 1HNU-HSF/LWL Collaborative Innovation Laboratory, School of Food Science and Engineering, Hainan University, 58 Renmin Road, Haikou 570228, China; helen_ht1205@163.com (T.H.); hnchwx@163.com (W.C.); hainufood88@163.com (Q.Z.); chenwj@hainu.edu.cn (W.C.); xyp18161011146@163.com (Y.X.); wujiawu1115@163.com (J.W.); 2Maritime Academy, Hainan Vocational University of Science and Technology, 18 Qiongshan Road, Haikou 571126, China

**Keywords:** zein, shellac, curcumin, edible film, functionality

## Abstract

The development of functional edible films is promising for the food industry, and improving the water barrier of edible films has been a research challenge in recent years. In this study, curcumin (Cur) was added to zein (Z) and shellac (S) to prepare an edible composite film with a strong water barrier and antioxidant properties. The addition of curcumin significantly reduced the water vapor permeability (WVP), water solubility (WS), and elongation at break (EB), and it clearly improved the tensile strength (TS), water contact angle (WCA), and optical properties of the composite film. The ZS–Cur films were characterized by SEM, FT-IR, XRD, DSC, and TGA; the results indicated that hydrogen bonds were formed among the curcumin, zein, and shellac, which changed the microstructure and improved the thermal stability of the film. A test of curcumin release behavior showed controlled release of curcumin from the film matrix. ZS–Cur films displayed remarkable pH responsiveness, strong antioxidant properties, and inhibitory effects on *E. coli*. Therefore, the insoluble active food packaging prepared in this study provides a new strategy for the development of functional edible films and also provides a possibility for the application of edible films to extend the shelf life of fresh food.

## 1. Introduction

Polyolefin plastics are widely used in food packaging materials, but degradation of most plastic packaging is very slow, and complete degradation requires decades or even hundreds of years, which causes great pollution in the environment [1]. To solve this problem, natural edible ingredients have been used as film-forming substrates to replace traditional plastic packaging, and this has become an important subject of scholarly research. Edible packaging film substrates are usually based on natural biological macromolecules such as proteins, polysaccharides, and lipids [2]. Therefore, in practical applications, the edible composite films keep materials fresh, edible, and pollutant-free, and they exhibit great application prospects [3].

Zein has great potential for producing edible packaging films due to its abundant raw materials, effective and biodegradable properties, and excellent water vapor barriers [4,5]. Because of the high content of hydrophobic amino acids, zein is insoluble in water but soluble in 50~95% alcohol and alkaline solutions with a pH greater than 11 [6]. In addition, zein films have poor mechanical properties, and plasticizers can be added to modify the zein film and improve its overall performance [7,8]. Previous studies have found that the addition of glycerol and polyethylene glycol to zein could improve the flexibility of the film [9]. In addition, adding appropriate amounts of oleic acid as a plasticizer to the zein solution could enhance the mechanical properties of the membrane [10], reduce the water vapor permeability, and improve the water resistance of the membrane [11].

Shellac is a natural hydrophobic polymer, and its main components are hydroxy fatty acids and sesquiterpene acids, which contain many carboxyl groups and carbonyl groups [12]. Shellac is nontoxic and physiologically harmless, meaning it can be used as a food additive (E904) [13]. In the fields of food and agriculture, shellac is commonly used to isolate water, gas, lipids, and microorganisms and to prolong the shelf lives of products [14,15,16,17]. In the pharmaceutical industry, it is used for moisture control with drugs, control systems, and the intestinal coatings of drugs and probiotics [18,19]. In research on edible shellac film materials, it has been found that shellac reduces the WVP (water vapor permeability) of edible films [20] and effectively extends the shelf lives of products [21]. Some researchers have tried to improve the barrier properties of shellac by adding zein [22].

Curcumin is a natural edible pigment that is widely used in various confectioneries, pastries, beverages, and other foods [23,24]. The hydroxyl groups at both ends of the curcumin molecule are conjugated with the electron cloud and cause deviation under alkaline conditions. The color of the curcumin solution changes from orange to red when the pH is greater than eight, so it can be used as a pH acid–base indicator based on this property [25,26]. In addition, curcumin has both antioxidant and antibacterial properties [27]. Curcumin can be added to edible packaging films as an indicator used to monitor changes in food quality and freshness [28].

To the best of our knowledge, no studies have been reported on composite films of zein and shellac loaded with curcumin. In this study, a new edible packaging film was prepared by blending these three natural components using the solution-casting method. The fundamental properties of the composite film such as mechanical, barrier, and optical properties were studied, and the morphological structure and thermal properties of the composite film were analyzed by scanning electron microscopy (SEM), Fourier transform infrared spectroscopy (FT-IR), X-ray diffraction (XRD), thermogravimetric analysis (TGA), and differential scanning calorimetry (DSC). In addition, curcumin was used as an antioxidant and colorimetric indicator in a composite film, and the functional properties (e.g., curcumin release ability, pH response, and antioxidant and antibacterial properties) of the composite film were investigated. The purpose of this study is to prepare edible packaging films with excellent barrier properties and functional activities and to provide theoretical support for the development and application of multifunctional food packaging.

## 2. Materials and Methods

### 2.1. Materials

Zein (protein content of 90%) was purchased from Sigma-Aldrich, (Steinheim, Germany). Shellac (chemical pure grade) was purchased from Sinopharm Chemical Reagent Co., Ltd. (Shanghai, China). Curcumin (analytical reagent grade) was purchased from Shanghai Yuanye Biotechnology Co., Ltd. (Shanghai, China). Tributyl citrate and oleic acid were of analytical grade and purchased from Macklin Biochemical Co., Ltd. (Shanghai, China). Anhydrous ethanol, glycerin, calcium chloride, and other analytical grade reagents were purchased from Xilong Chemical Co., Ltd. (Guangzhou, China). 2,2-Diphenyl-1-picrylhydrazyl (DPPH) and 2,2′-azi-nobis (3-ethylbenzothiazoline-6-sulfonic acid) (ABTS) were purchased from Grace Biotechnology Co., Ltd. (Suzhou, China).

### 2.2. Preparation of Films

The process for composite film preparation was as follows: zein powder (2.0 g) was dissolved in 100 mL of ethanol solution (90%, *v*/*v*), heated and stirred by a magnetic stirrer (85-1, Aowa Instrument Co., LTD., Changzhou, China) at 65 ℃ for 30 min, and cooled to room temperature. The shellac powder (2.0 g) was dissolved in 100 mL of ethanol solution (90%, *v*/*v*) and stirred for 60 min at room temperature until the powder was completely dissolved. Subsequently, the two ethanol solutions were blended in equal volumes, and tributyl citrate (30%, *w*/*w*, based on the total solid mass of zein and shellac) was added. The mixtures were heated and stirred at 70 °C for 30 min. During cooling, oleic acid (25%, *w*/*w*, based on the total solid mass of zein and shellac) was added and homogenized at high speed by a high-speed shear homogenizer (T18, IKA Co., Ltd., Stauffen, Germany) operated for 2 min at 12,000 rpm.

The mixed solution was then degassed in an ultrasonic cleaner (SK5200HP, Keyguide Ultrasonic Instrument Co., Ltd., Shanghai, China) for 30 min. All accurately weighed film solutions were cast onto polystyrene Petri dishes, dried at 40 °C for 5 h, and finally peeled off. All films were tested after equilibrating for 48 h in the chosen environment (25 ± 2 °C and 50 ± 5% relative humidity).

### 2.3. Preparation of Curcumin-Loaded Films

Based on the optimal formula obtained from the single-factor (Appendix A) and orthogonal experiments (Appendix A), curcumin (Cur) was added to improve the water resistance and functional properties of the composite films. Accurately weighed curcumin was added into the mixed solution from Section 2.2., and it was stirred for 1 h in the dark. Then the mixed solution was degassed in an ultrasonic cleaner and dried at 40 °C for 5 h. The concentrations of Cur were 0% (as a control), 1%, 3%, 5%, and 7% (*w*/*w*). All films were stored in the chosen environment (25 ± 2 ℃, 50 ± 5% relative humidity) in the dark until further testing. The film numbers were ZS–Cur0, ZS–Cur1, ZS–Cur3, ZS–Cur5 and ZS–Cur7.

### 2.4. Physical Properties of the Films

#### 2.4.1. Film Thickness

Eight points were randomly selected on the films to measure the thickness with a digital thickness gauge (32CHQF1030, Deqing Shengtaixin Electronic Technology Co., Ltd., Huzhou, China) with a precision of 0.001 mm, and the results were given as the average values.

#### 2.4.2. *WVP*

The *WVP* of the film was measured according to a previous method [29,30], with some modifications. Briefly, the film samples were dried in a drying oven (DGG-9123A, Senxin Experimental Instrument Co., Ltd., Shanghai, China) at 120 °C for 2 h. Then, the films were cut into squares measuring approximately 40 × 40 mm, and weighing bottles (depth 2.5 cm, exposed area 12.56 cm^2^) filled with 3 g of anhydrous calcium chloride (0% RH) were sealed with square film. After determining the initial weight, the sample was placed in a glass desiccator with 90 ± 3% RH for 48 h at room temperature, and its weight was recorded every 24 h. The *WVP* of the film was calculated with the following formula:(1)$$WVP=\frac{△m\times d}{A\times t\;\times △P}$$
where △m is the weight change (g), d is the film thickness (mm), A is the permeation area (m^2^), t is the permeation time (h), △P is the water vapor pressure difference (kPa) between the two ends of the film, and *WVP* is expressed in units of g⋅mm⋅m^−2^⋅h^−1^⋅kPa^−1^. Each film sample test was repeated four times under the same conditions.

#### 2.4.3. *WS* (Water Solubility)

The *WS* of the film was defined as the percentage of the dry matter of the film dissolved after immersion in water for 24 h [31]. The *WS* of the film was measured according to the method of Khoshgozaran-Abras et al. [32], with some modifications. The films were cut to 20 × 20 mm sections, dried at 40 °C for 24 h, and weighed as *M*_1_. The dried samples were subsequently immersed in a conical flask containing 100 mL of distilled water and shaken at 100 rpm for 24 h at room temperature. The undissolved films were collected by vacuum filtration and dried at 40 °C until a constant weight was reached and weighed as *M*_2_. Each group of samples was tested four times under the same conditions. The *WS* was calculated as follows:(2)$$WS\left(\%\right)=\frac{M_{1}-M_{2}}{M_{1}}\times 100$$

#### 2.4.4. Mechanical Properties

*TS* and *EB* of the films were measured with a tensile testing machine (3343, INSTRON, Norwood, MA, USA) according to the ASTM D882 standard method [33,34]. Film samples were cut into strips measuring 10 × 70 mm and placed in a desiccator at 50 ± 5% RH for 48 h at room temperature. The initial grip separation distance of the cross probe was set to 50 mm, and the pulling speed was set to 1.0 mm/s. The maximum tensile force at break was recorded as *F_max_* (N), the cross-sectional area was *S* (mm^2^), the initial length of the film was *L*_0_ (mm), and the length at break was *L*_1_ (mm). Each group of samples was tested four times under the same conditions, and *TS* and *EB* were calculated with the following formulas:(3)$$TS\;\left(MPa\right)=\frac{F_{max}}{S}$$
(4)$$EB\;\left(\%\right)=\frac{L_{1}-L_{0}}{L_{0}}\times 100\%$$

#### 2.4.5. Optical Properties

The colors of the films were measured with a colorimeter (WSC-1B, Yi Electric Physical Optical Instrument Co., LTD., Shanghai, China). *L** (lightness), *a** (redness–greenness) and *b** (yellowness–blueness) were used as the color parameters of the films, and a standard white plate ($L_{0}^{*}\;$= 96.59, $a_{0\;}^{*}$= −0.13, $b_{0}^{*}$ = −0.11) was used as a background reference. The total color difference (Δ*E*), whiteness index (*WI*), and yellowness index (*YI*) were calculated using the following equations [35]:(5)$$\Delta E=\sqrt{{\left(L^{*}-L_{0}^{*}\right)}^{2}+{\left(a^{*}-a_{0}^{*}\right)}^{2}+{\left(b^{*}-b_{0}^{*}\right)}^{2}}$$
(6)$$WI=100-\sqrt{{\left(100-L^{*}\right)}^{2}+{\left(a^{*}\right)}^{2}+{\left(b^{*}\right)}^{2}}$$
(7)$$YI=\frac{142.86b^{*}}{L^{*}}$$
where $L^{*}$, $a^{*}$, and $b^{*}$ are the color values of the samples, and $L_{0}^{*}$, $a_{0}^{*}$, and $b_{0}^{*}$ are the color values of the standard white plate. The value 142.86 is the constant, and the *YI* calculation formula refers to Farajpour et al. [35]. The measurements were repeated five times for each sample.

The opacities of the films were measured with an ultraviolet-visible spectrophotometer (TU1810, Putuo General Equipment Co., Ltd., Beijing, China), and the absorbance values (*A*_600_) of the samples were recorded at 600 nm and at room temperature. The films were cut into rectangles measuring 10 × 40 mm and fixed on one side of the cell. An empty cell was applied as a reference. The *opacity* was calculated as follows [36]:(8)$$Opacity=\frac{A_{600}}{d}$$
where $A_{600}$ is the absorbance at 600 nm and d is the film thickness (mm). The measurement was repeated five times for each sample.

To check the light-blocking effect of the film against UV radiations, the transmittance of film samples was measured in the wavelength range of 200~800 nm using air as the reference. After irradiation by UV for 12 h, the transmittance of the film was measured at 300 nm to analyze the transmittance change of the film under prolonged UV irradiation.

#### 2.4.6. WCA

The contact angle of the films was studied with a video optical contact angle measuring instrument (OCA15EC, DataPhysics Instrument Co., Ltd., Filderstadt, Germany) using the sessile method at room temperature [37]. A film (20 × 20 mm) was fixed on a glass slide, and deionized water (5 μL) was slowly dripped on the surface of the films with a 1 mL syringe supplied with the instrument. The WCA was measured immediately and again after 60 s. The captured images and calculations of the angles for the water drop contact surface on both sides were analyzed with SCA20 (DataPhysics Instrument, Charlotte, NC, USA) software.

### 2.5. Characterization of the Films

#### 2.5.1. SEM

The surface and cross-sectional micromorphologies of the films were investigated with SEM (S-3000N, Hitachi Co., Ltd., Tokyo, Japan) at an acceleration voltage of 10 kV, and the images were photographed. For cross-sectional analyses, the obtained films were fractured in liquid nitrogen to reduce the damage caused by external forces and to obtain a flat cross-section. Furthermore, all films were coated with gold before observation. Images with the magnification of 1.0 k (surface) and 3.0 (cross-section) were recorded.

#### 2.5.2. FT-IR

Functional group changes of the films were analyzed by FT-IR (Tensor27, Bruker Optics Co., LTD., Ettlingen, Germany). Before testing, the film samples were dried at 40 °C for 24 h, and then the samples (1–2 mg) were mixed thoroughly with dried potassium bromide (200 mg) in a 1:100 ratio and pressed into thin wafers. The scanning background was air, the scanning range was 4000 to 400 cm^−1^, and the films were scanned 32 times with a resolution of 4 cm^−1^. After denoising and baseline correction, the absorption spectrum of the film was obtained.

#### 2.5.3. XRD

An X-ray diffractometer (XRD, Smart Lab, Rigaku Co., Ltd., Tokyo, Japan) was applied to record XRD data for the films. An accelerating voltage of 40 kV was applied at 30 mA using Cu-Kα radiation (λ = 1.5418 Å) with a scan range of 5 to 90° (2θ) and a scan rate of 5°/min.

#### 2.5.4. DSC

The shellac powder, zein powder, and composite film samples were analyzed by DSC (Q100, TA Instruments Co., LTD., Lukens Drive New Castle, DE, USA). A 4–5 mg sample was weighed into an aluminum pot, sealed, and heated over the temperature range 20–200 °C with a heating rate of 10 °C/min and a nitrogen flow rate of 50 mL/min. An empty aluminum pot was used as a blank control.

#### 2.5.5. TGA

The thermal properties of the films were analyzed by TGA (Q600, TA Instruments Co., LTD., Lukens Drive New Castle, DE, USA) under a nitrogen atmosphere with a flow rate of 50 mL/min and a heating rate of 10 °C/min from 25 °C to 600 °C. A 10 mg sample was accurately weighed into an aluminum pot for testing. All samples were stored in a dry environment before testing.

### 2.6. Curcumin Release Tests

The semi-fatty food simulant medium (50% ethanol, *v*/*v*) and fatty food simulant medium (95% ethanol, *v*/*v*) were employed as models to evaluate the release of curcumin from the ZS–Cur films [38,39]. Film samples (20 mm × 20 mm) were immersed into 200 mL of the corresponding food simulant and stored in the dark at room temperature for 7 days, and the shaking speed was set at 100 rpm. Four milliliters were removed from the simulant at set periods, and the absorbance values at 470 nm were measured. After each measurement, the testing solution was immediately poured back into the original solution to keep the total volume of the solution constant. The concentration of curcumin at the corresponding time was calculated from the standard curve. The cumulative release rate (%) of curcumin was expressed as the percentage of curcumin released at different times relative to the total amount of curcumin in the film [3]. The accumulative release rate (%) was calculated as follows:(9)$$Accumulative\;release\;\left(\%\right)=\frac{M_{t}}{M_{0}}\times 100\%$$
where $M_{t}$ is the amount of curcumin released at time *t* and $M_{0}$ is the total amount of curcumin in the film.

### 2.7. pH Response Tests

Curcumin powder was separately dissolved in different pH phosphate buffer solutions (pH 3–7, 9–11) at the same final concentration (0.1 mg/mL), and a buffer solution without curcumin was used as a blank control. A digital camera was used to record the color changes of the solutions.

The prepared ZS–Cur film (20 mm × 20 mm) was separately immersed in the different pH phosphate buffers (pH 3–7, 9–11) for 5 min. After removing the surface water of the film with filter paper, the color parameters and visible color changes of the films were determined with a colorimeter and a digital camera. The color parameters of the films were shown by *L** (lightness), *a** (redness–greenness), and *b** (yellowness–blueness), and $\Delta E_{1}$ was calculated with the following formula:(10)$$\Delta E_{1}=\sqrt{{\left(L_{2}^{*}-L_{1}^{*}\right)}^{2}+{\left(a_{2}^{*}-a_{1}^{*}\right)}^{2}+{\left(b_{2}^{*}-b_{1}^{*}\right)}^{2}}$$
where $L_{2}^{*}$, $a_{2}^{*}$, and $b_{2}^{*}$ are the color values of the samples, and $L_{1}^{*}$, $a_{1}^{*}$, and $b_{1}^{*}$ are the color values of the standard white plate ($L_{1}^{*}\;$= 95.37, $a_{1}^{*}\;$= −0.76, and $b_{1}^{*}\;$= 5.20).

### 2.8. Antioxidant Properties

#### 2.8.1. *DPPH* Radical Scavenging Activity

The *DPPH* radical scavenging activity was measured according to the method of Xiao et al. [3]. Briefly, the film sample (5 mg) was immersed in 10 mL of ethanol (95%, *v*/*v*) and stirred in the dark for 12 h at room temperature. Then the supernatant was collected for testing by centrifugation at 4000× *g* for 10 min at 4 °C. The supernatant was diluted and reacted with the *DPPH* working solution (150 µL) in the dark for 30 min at room temperature. Then, the absorbance values at 517 nm were determined with a microplate reader (Synergy LX, Bio Tek Instrument Co., Ltd., Winooski, VT, USA). The sample solution was mixed with an 80% methanol solution for the control group, and an 80% methanol solution was mixed with *DPPH* solution for the blank group; the remaining steps were the same. The *DPPH* radical scavenging rates of the composite films were calculated as follows:(11)$$DPPH\;radical\;scavenging\left(\%\right)=\left(1-\frac{A_{i}-A_{c}}{A_{0}}\right)\times 100$$
where $A_{i}$ is the absorbance value of the measured sample, $A_{c}$ is the absorbance value of the control group, and $A_{0}$ is the absorbance value of the blank group.

#### 2.8.2. *ABTS* Radical Scavenging Activity

The *ABTS* radical scavenging activity was measured with a previous method [40]. The extracted film solution was reacted with the *ABTS* working solution (190 µL) in the dark for 6 min at room temperature, and the absorbance value at 734 nm was determined immediately. The sample solution was mixed with anhydrous ethanol for the control group, and anhydrous ethanol was mixed with the *ABTS* solution for the blank group; the remaining steps were the same. The *ABTS radical scavenging* rates of the composite films were calculated as follows:(12)$$ABTS\;radical\;scavenging\left(\%\right)=\frac{A_{i}-A_{c}}{A_{0}}\times 100$$
where $A_{i}$ is the absorbance value of the measured sample, $A_{c}$ is the absorbance value of the control group, and $A_{0}$ is the absorbance value of the blank group.

### 2.9. Antibacterial Properties

Escherichia coli (*E. coli*), a common Gram-negative bacterium, was selected as the test bacterium, and the antibacterial activities of the ZS–Cur films were measured by the bacteriostatic circle method. For the antimicrobial test method, readers are referred to Huang et al. [41]. The prepared plate count agar (PCA) solution (25 mL) was poured into a sterile medium, solidified after being sterilized and cooled, inoculated with 0.5 mL of the strain suspension (1 × 10^7^ CFU/mL), and evenly coated. The composite films were made into discs with diameters of 6 mm with a hole punch. After sterilization by UV irradiation for 30 min, the discs were attached to the inoculated medium. The blank group was filter paper soaked in sterile water. All of the above operations were carried out on an ultraclean bench (SW-CJ-1FD, Jiabao Purification Engineering Equipment Co., LTD., Suzhou, China), and each group of samples was prepared three times under the same conditions. Then, the media were inverted in a constant-temperature incubator (SPX-288, Jiangnan Instrument Factory, Ningbo, China) at 37.5 °C for 24 h, and the antibacterial diameters were measured by the cross method. The larger the diameter of the antibacterial ring, the better the antibacterial effect.

### 2.10. Statistical Analyses

These experiments were repeated at least in triplicate, and the data were expressed as the mean ± standard deviation. Statistical analyses were accomplished by one-way analysis of variance (ANOVA) using SPSS 23 software (IBM Inc., Armonk, NY, USA), and a significant difference was defined as a *p*-level of 0.05 (LSD test).

## 3. Results and Discussion

### 3.1. Physical Properties of ZS–Cur Films

#### 3.1.1. WVP, WS, TS, and EB Analyses

The WVP, WS, TS, and EB values of ZS films with different curcumin contents are shown in Table 1. The water vapor barrier properties of the composite films were affected by the curcumin content (*p <* 0.05), and the WVP values gradually decreased as the curcumin content was increased from 0 to 3%. This indicated that the addition of curcumin enhanced the water vapor barrier properties of ZS–Cur films, which is likely because the rod-shaped crystals of curcumin that are uniformly dispersed in the film substrates blocked water vapor penetration [25]. The improved barrier capacities could also be explained by the formation of hydrogen bonding interactions between curcumin, zein, and shellac [42], and the benzene rings and long carbon chains in the chemical structure of curcumin were hydrophobic, which enhanced the water vapor barrier properties of the ZS–Cur film [43]. As the curcumin content was increased further to 5% and 7%, the water vapor barrier performance of the films gradually decreased because the uneven dispersion of excessive aggregated curcumin in the film substrates and disruption of the dense structure in the films increased the WVP values. However, the WS values of ZS–Cur films decreased slightly with increasing curcumin content, and the overall difference was not significant (*p >* 0.05).

Furthermore, the addition of curcumin significantly enhanced the TS of ZS–Cur films (*p <* 0.05). With increased addition of curcumin, the TS of ZS–Cur films increased and then decreased, and the TS values of the films were the highest when the curcumin content was 3%. However, the EB values decreased significantly as the curcumin content was increased (*p <* 0.05). The low curcumin content was uniformly dispersed in the film substrate, and it had a large number of hydroxyl groups that formed many hydrogen bonds with the film substrates and increased the TS. At the same time, due to the enhancement of hydrophobicity, the storage humidity of the film was reduced, which leads to a decrease in EB value [44,45].

#### 3.1.2. Optical Properties

The color and opacity results for ZS films loaded with different curcumin contents are shown in Table 2. The addition of curcumin had a significant effect on the L*, a*, and b* values of ZS–Cur films (*p <* 0.05). The L* values of ZS–Cur films were significantly lower than those of the films without curcumin. Xiao et al. [3] also concluded that the addition of curcumin to a cellulose film reduced L* values. As the curcumin content was increased, the L* values of the films gradually decreased, while the a* and b* values showed increasing trends, and the ∆E and YI values also showed increasing trends. However, ZS–Cur films showed significantly lower WI values than the ZS films without curcumin (*p <* 0.05). The above findings suggested that the color characteristics of the ZS films could be adjusted by changing the amount of curcumin.

Compared to the ZS films without curcumin, the ZS films containing curcumin showed higher opacities (*p <* 0.05), and the opacities of the ZS–Cur films increased significantly with increasing curcumin content (*p <* 0.05). Perhaps curcumin, which is an active photosensitive compound, absorbed and scattered light and effectively reduced the light transparencies of the films [46]. 

Figure 1A shows the UV-Vis transmission spectra of films with different Cur contents. As can be seen, all films showed excellent UV-blocking performance (200–400 nm). In the visible region, the films containing Cur showed low light transmittance, which is mainly caused by the absorption and scattering of light by curcumin with a rod-like crystal structure [25,46]. Figure 1B shows the change of UV transmittance of the films under 12 h ultraviolet irradiation. With the extension of irradiation time, the overall transmittance of the film first increased and then decreased significantly, which probably related to the photosensitive reaction of curcumin. The above study showed that the prepared ZS–Cur films had good light-blocking properties and potential for protecting food from oxidative deterioration [3].

#### 3.1.3. WCA

The WCA was used as an index to reflect the surface hydrophobicity of the packaging films, which is not only related to the chemical properties of the film but is also affected by the surface microstructure [47]. Table 3 shows the WCA values of ZS–Cur films measured at 0 s and 60 s. The WCA values of ZS-based films were less than 90°, which was not expected based on the hydrophobicity of zein and shellac. This was likely influenced by the surface roughness of the films [48]. In addition, during the process of solvent evaporation and solute solidification, the hydrophobic interactions and self-aggregation behavior of the zein molecules resulted in primary distribution of the hydrophilic groups of the macromolecules on the surfaces of the film substrate [48,49]. In particular, there was no significant difference between the WCA values of ZS–Cur1 films and ZS–Cur0 films at 0 s and 60 s (*p >* 0.05). With increasing curcumin content, the WCA values of the composite films tended to increase, indicating that the hydrophobicity of curcumin played an important role. The WCA values decreased after 60 s for all film samples. ZS–Cur0 and ZS–Cur1 had similar WCA values, and the WCA values of ZS–Cur3, ZS–Cur5, and ZS–Cur7 were 23.23°, 25.47°, and 36.43°, respectively [37]. The ∆WCA values of the ZS films gradually decreased as the curcumin content was increased, and it was obvious that the rod-shaped crystalline structure of curcumin hindered permeation by the water droplets on the film surface.

### 3.2. Characterization of ZS–Cur Films

#### 3.2.1. SEM

Figure 2 contains SEM images of the surfaces and cross-sections of ZS films with different curcumin contents, where Figure 2a–e are surface views of ZS–Cur0, ZS–Cur1, ZS–Cur3, ZS–Cur5, and ZS–Cur7, respectively, and Figure 2A–E show the cross-sectional views. The addition of curcumin affected the micromorphological structures of the films. The dense morphologies displayed on both the surfaces and cross-sections of the ZS films without curcumin were due to the good film-forming ability of zein and shellac and their strong bonding during the formation process. However, a small number of particles appeared in the cross-sectional views, which could be attributed to rapid evaporation of the ethanol during the drying process, which led to enhanced hydrophobic interactions between zein and shellac molecules and a higher degree of macromolecular aggregation [50,51].

Compared to the ZS–Cur0 film, the surfaces of the films incorporating curcumin were rougher, while the cross-sections showed higher denseness. However, at lower levels (1% and 3%), curcumin was better distributed in the film substrates, which was beneficial in enhancing the mechanical and barrier properties of the films. However, with further increases in the curcumin content (5% and 7%), the uniformity of the composite film worsened, and the film surface became gradually rougher with more particles and a looser cross-sectional structure. This suggested that as the solvent evaporated during drying, excess curcumin crystals were aggregated and randomly accumulated in the film substrate [25].

#### 3.2.2. FT-IR Spectroscopy

FT-IR was used to investigate the interactions of functional groups in the composite films. As shown in Figure 3A, the FT-IR spectrum of pure shellac demonstrated a broad band at 3450.58 cm^−1^, which was assigned to O-H stretching; a peak at 2934.59 cm^−1^ corresponding to C-H stretching; and a peak at 1735.43 cm^−1^ corresponding to C = O stretching [52]. The peak at 723.74 cm^−1^ was assigned to olefin C-H out-of-plane bending. Zein had two characteristic amide band regions, of which the amide I region (1680–1630 cm^−1^) indicated the secondary structure of the protein, such as an α-helix or β-fold, and the amide II region (1655–1590 cm^−1^) indicated the presence of hydrogen bonds [53]. Pure zein showed a characteristic peak at 1656.31 cm^−1^ in the amide I region for C = O stretching and another peak at 1539.91 cm^−1^ in the amide II region for bending of N-H groups and stretching of C-N groups [54,55]. Meanwhile, a broad band at 3323.39 cm^−1^ in the amide A region was associated with O-H and N-H stretching [56], and a peak at 2926.90 cm^−1^ in the amide B region was assigned to C-H stretching [57].

The FT-IR spectra of ZS–Cur0 films showed that the peaks for the O-H and C-H groups were shifted to 3416.63 and 2929.40 cm^−1^, respectively, indicating hydrogen bonding interactions between the shellac and zein. However, the absorption peak in the amide II region shifted to 1542.38 cm^−1^, suggesting that there were also hydrophobic interactions between the zein and shellac, which were due to the high content (>50%) of hydrophobic amino acids in zein and hydrophobic groups in shellac [58,59]. As curcumin was added to the composite films, the characteristic peaks for both shellac and maize alcoholic protein appeared in the FT-IR spectra of the composite films. The FT-IR spectrum of the ZS–Cur1 film was similar to that of the ZS–Cur0 film, and as the curcumin content was gradually increased, the position of the peak corresponding to N-H groups shifted, and the peak intensity gradually decreased, which indicated that N-H groups content gradually decreased [33]. As seen in Figure 3B, with increasing curcumin content, a new peak appeared between 1550~1500 cm^−1^, 1514.80 cm^−1^, and 1513.53 cm^−1^ associated with C = C stretching of the benzene rings, which is characteristic of curcumin [60,61]. The above results suggested that molecular interactions between the C = C groups of curcumin and the film substrates occurred when the added curcumin content was low (1% and 3%), while the characteristic peaks for the C = C group appeared in the FT-IR spectra of ZS–Cur films with excess curcumin (5% and 7%) [33]. In addition, a new peak was found at 1190.59~1189.04 cm^−1^ in the composite film spectrum for the antisymmetric C-O-C stretching [29,62], and a peak at 724.22~723.90 cm^−1^ was assigned to a methylene CH_2_ in-plane wobble vibration, which was associated with the antisymmetric stretching of alkane C-H at 2930 cm^−1^.

#### 3.2.3. XRD

The degree of crystallization of a substance can be judged from the XRD pattern. Substances with sharp peaks are regarded as crystalline, while those without obvious peaks are regarded as amorphous [63]. As seen in Figure 4A, the zein powder showed two diffraction peaks at 2θ = 9.26° and 19.74°, and the shellac powder had a broad peak at 2θ = 18.75°. These results indicated that zein and shellac adopted typical amorphous structures, which was consistent with the results of other studies [64,65,66]. In previous studies, the XRD pattern for curcumin powder showed multiple narrow and sharp crystalline peaks, indicating that curcumin had a crystalline structure [63]. In addition, all films showed main diffraction peaks at approximately 2θ = 20°, and with increased curcumin content, the peak width showed no obvious change while the peak intensity decreased slightly, which suggested that the films were amorphous structures with sequentially decreasing crystallinity [67]. Moreover, a comparison of the XRD patterns for the five ZS films showed no significant differences. All the films showed similar peak shapes, and there were no characteristic peaks for curcumin and no additional strong peaks in the XRD patterns of the films. These findings pointed to a physical connection between curcumin and the film substrates, and the addition of curcumin did not change the crystalline structures of the films. The related literature has reported similar results, such as with antioxidant composite films based on curcumin nanocapsules, polyvinyl butyral films loaded with curcumin, and chitosan coatings loaded with curcumin [3,63,68].

#### 3.2.4. Thermal Performance Analyses

DSC curves for shellac, zein, and the composite film are shown in Figure 4B. The zein powder and shellac powder showed broad endothermic peaks at 91.08 °C and 63.71 °C, respectively, which could be related to evaporation of bound water from the polymer [59,69]. The DSC curve for zein showed a decreasing trend when the temperature was increased beyond 220 °C, which approached the decomposition temperature of zein [70]. However, no decomposition was observed in the DSC curve of shellac, indicating that shellac was thermally stable in the 250 °C range [58]. A small endothermic peak was observed at 59.33 °C in the DSC curve of the ZS–Cur0 film, which was probably due to the plasticizing effects of the added tributyl citrate and oleic acid on the film substrates, which kept the temperature of the ZS–Cur0 film lower than those of shellac and zein at this stage [71]. A broad endothermic peak appeared at 126.26 °C, which may be the melting point of the ZS–Cur0 film [72]. As the temperature continued to rise above 180 °C, the ZS–Cur0 film began to decompose [73]. In addition, the DSC curves for ZS–Cur1 and ZS–Cur3 films showed similar trends, with the first endothermic peaks at 85.88 °C and 74.63 °C, respectively, which were higher than those for the ZS–Cur0 film. As the temperature continued to increase, second endothermic peaks appeared at 231.98 °C and 219.86 °C for ZS–Cur1 and ZS–Cur3 films, respectively, presumably due to melting of the composite films, and no decomposition of the films was observed at this stage. However, the ZS–Cur5 and ZS–Cur7 films showed first endothermic peaks at 76.67 °C and 71.04 °C, respectively. When the temperature continued to increase beyond 180 °C, the ZS–Cur5 and ZS–Cur7 films were decomposed at 185.17 °C and 191.66 °C, respectively. These results suggested that curcumin effectively improved the thermal stabilities of the composite films when its content was low. High contents of curcumin were unevenly distributed in the films, which disrupted the film continuity and resulted in poor thermal stability.

The TGA and DTG curves of shellac, zein, and films are shown in Figure 4C,D. The weight loss peak temperature (°C), weight loss %, and residue % of the samples are shown in Table 4. Shellac and zein showed significant weight loss peaks at 438.89 °C and 328.82 °C, respectively, with weight losses of 65.54% and 38.37%. The TGA curves showed that the thermal decomposition temperatures of shellac and zein were 280~520 °C and 230~420 °C, respectively, and the residue of zein was higher (17.658%) while that of shellac was lower (8.033%) at 600 °C. The first weight loss of the ZS films occurred at 150~280 °C and was probably due to decomposition of the plasticizer tributyl citrate [5]. The second weight loss peak occurred between 280 and 360 °C and was caused by breakage of zein molecular chains and thermal decomposition [74,75]. The third stage of weight loss involved thermal pyrolysis of the shellac, which occurred between 360 and 530 °C [2]. Table 4 shows that the second and third peaks of the ZS–Cur3 film exhibited the lowest weight losses. The final residues of the films constituted between 8.033% and 17.658% of the initial weights, and the residues of the ZS–Cur3, ZS–Cur5, and ZS–Cur7 films were significantly higher than that of the ZS–Cur0 film. The above results indicated that the addition of curcumin significantly enhanced the thermal stabilities of ZS films.

### 3.3. Curcumin Release Properties

The data for release of Cur from Cur-containing films into semi-fatty and fatty food simulants are shown in Figure 5A,B. All films in both simulation systems exhibited similar release profiles, i.e., initial bursts in the early stages and subsequent sustained release until reaching equilibrium, which indicated that ZS–Cur films controlled curcumin release effectively [3,25]. The initial burst was likely caused by rapid release of curcumin from and near the film surface [25]. As expected, the initial release rate and accumulative release of curcumin in the fatty food simulation were slightly higher than those in the semi-fatty food simulation due to the lipophilic nature of curcumin and the stability differences of the various film structures used in each simulation [38].

Moreover, the higher curcumin contents of ZS–Cur films required longer release times to reach equilibrium, which was likely due to good dispersion and the embedded structure of curcumin in the films enabling well-controlled release. Compared with ZS–Cur5 and ZS–Cur7 films, ZS–Cur1 and ZS–Cur3 films showed higher cumulative release rates during the equilibrium phase. It was assumed that the high contents of curcumin caused self-aggregation in the films, distribution of hydrophilic groups on the film surfaces, and aggregation of the lipophilic groups in the films, thus hindering outward movement of the curcumin within the films [48,76]. These results suggested that the Cur-loaded ZS films underwent slow release and were beneficial as food packaging used to extend shelf lives [31,77].

### 3.4. pH Response of ZS–Cur Films

The colors of curcumin solutions at different pH values are shown in Figure 5C. The curcumin solutions were bright yellow at pH 3~7, and they gradually turned reddish brown with increasing alkalinity (pH 9~11). The color changes are attributed to reversible structural conversions shown by curcumin at different pH values [78].

Similarly, ZS–Cur films also exhibited visible color changes when immersed in phosphate buffers with different pH values, as shown in Table 5. The colors of all four films (ZS–Cur1, ZS–Cur3, ZS–Cur5, and ZS–Cur7) changed significantly as the curcumin content was increased. At pH values of 3~7, the colors of the four films changed from bright yellow to dark yellow, and the corresponding L* and b* values were significantly higher than those seen under alkaline conditions (*p <* 0.05). At a pH of 9, the a* values increased significantly (*p <* 0.05), and the film colors changed from yellow to orange-red. Then, the a* values gradually increased with increasing alkalinity, which suggested that a redshift occurred under alkaline conditions. This was because the keto form of the curcumin molecular structure dominates under acidic and neutral conditions [79], while the enol form predominates under alkaline conditions when curcumin undergoes degradation reactions [80,81].

### 3.5. Antioxidant Properties of ZS–Cur Films

Antioxidant activity is an important functional property of composite packaging materials and can be evaluated by measuring DPPH and ABTS radical scavenging with composite films. As shown in Figure 6A,B, ZS–Cur0 films exhibited low antioxidant activities, with DPPH and ABTS radical scavenging rates of 20.89% and 32.86%, respectively, probably due to the antioxidant properties of zein in the films, which was consistent with findings reported by others [64]. The antioxidant activities of films with added curcumin were significantly higher than those of films without curcumin (*p <* 0.05); with increasing curcumin content (1~7%), the DPPH radical scavenging rate increased from 26.57% to 80.46%, and the ABTS radical scavenging rate increased from 36.02% to 69.67%. The antioxidant activity of curcumin is mainly derived from the phenolic hydroxyl groups in its structure [82]. In addition, the composite films showed higher scavenging activities for DPPH radicals than for ABTS radicals, which was due to the different detection systems applied with DPPH and ABTS [83,84].

### 3.6. Antibacterial Properties of ZS–Cur Films

Bacterial proliferation is one of the main causes of food spoilage. Therefore, antibacterial properties are important indicators for evaluating food packaging materials with natural active ingredients [85]. In this experiment, *E. coli* was selected to investigate the antibacterial properties of shellac, zein, and films, as shown in Figure 5D. The pure shellac film and zein film did not show any antibacterial activity, while the ZS–Cur0 film showed obvious antibacterial activity against *E. coli*, which was presumably due to the dense structure formed by the zein and shellac composite. The ZS–Cur1, ZS–Cur3, and ZS–Cur5 films showed significant bacteriostatic effects (*p <* 0.05), and the antibacterial activities improved with increased curcumin addition. This indicated that successful release of curcumin from the ZS–Cur films enhanced their antibacterial activities. The ZS–Cur7 film had a smaller diameter for the antibacterial circle, which corresponded to the results described for curcumin release in Section 3.6. The antibacterial properties of curcumin were attributed to its ability to disrupt cell membranes, thereby releasing cytoplasm, disrupting organelles, and inhibiting normal cellular activities [86].

## 4. Conclusions

In this study, new zein/shellac-based composite films loaded with curcumin (ZS–Cur) were prepared by the solution casting method, and they showed good water barrier properties, pH responses, and antioxidant and antibacterial activities. Compared with ZS film without curcumin, the addition of curcumin significantly improved the water barrier properties, tensile strengths, opacities, and thermal stabilities of the films. Microstructural studies indicated that the denseness of the film was slightly reduced by adding curcumin. Moreover, the ZS–Cur films effectively controlled the release of curcumin and displayed good color responses to pH. The addition of Cur made the composite films display strong antioxidant activities and inhibition of *E. coli*. Therefore, the insoluble active food packaging prepared in this study provide a new strategy for developing new functional food packaging films. In a further study, practical applications of the ZS–Cur films will be verified with fresh food detection experiments.

## Figures and Tables

**Figure 1 foods-12-01577-f001:**
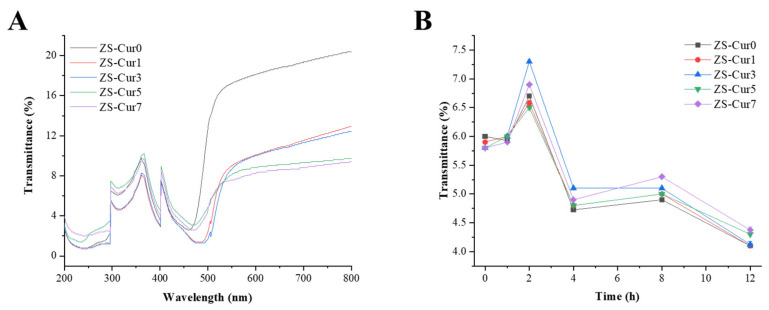
(**A**) UV-Vis transmittance of ZS–Cur films and (**B**) effect of prolonged UV irradiation on films.

**Figure 2 foods-12-01577-f002:**
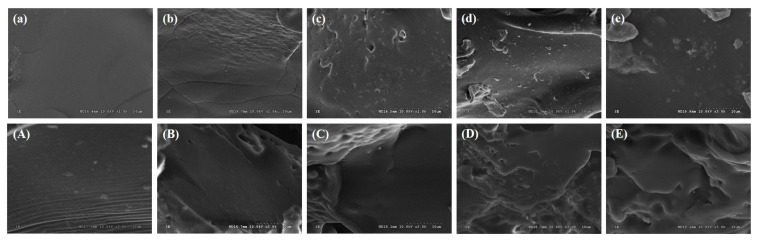
SEM images of ZS–Cur0 (**a**,**A**), ZS–Cur1 (**b**,**B**), ZS–Cur3 (**c**,**C**), ZS–Cur5 (**d**,**D**), and ZS–Cur7 (**e**,**E**). (**a**–**e**) demonstrate the surface area images, and (**A**–**E**) demonstrate the cross-section images.

**Figure 3 foods-12-01577-f003:**
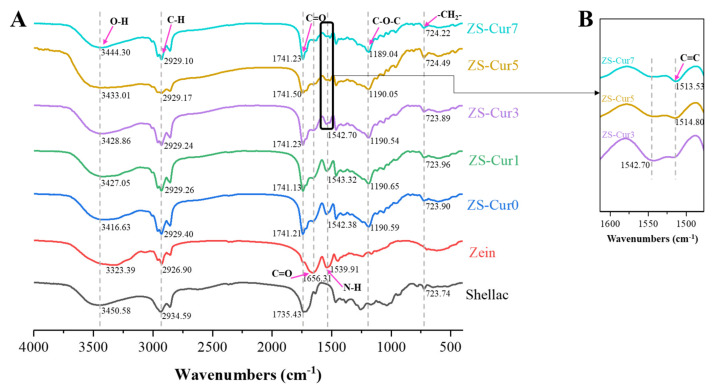
FT-IR spectra of zein/shellac films loaded with curcumin; (**A**)—FT-IR spectrum of pure shellac curcumin; (**B**)—FT-IR spectrum of pure shellac with increased curcumin content.

**Figure 4 foods-12-01577-f004:**
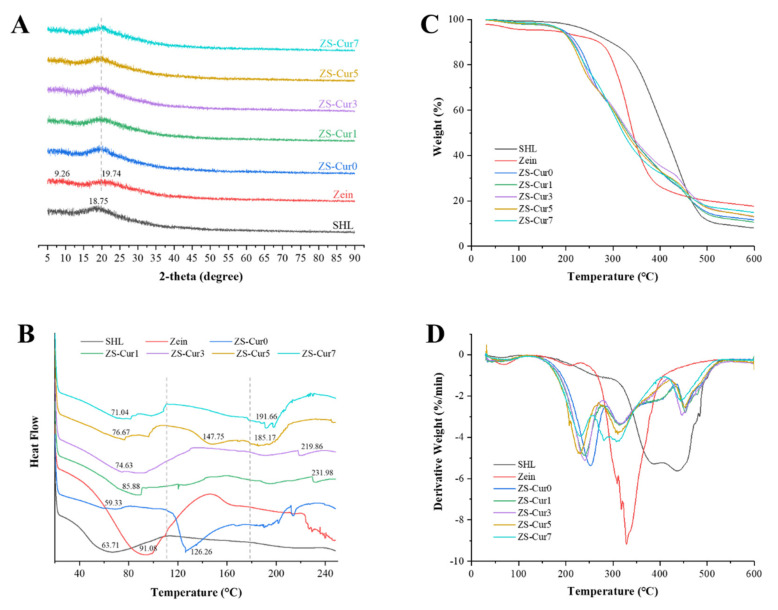
(**A**) XRD patterns, (**B**) DSC curves, (**C**) TGA, and (**D**) DTG curves of zein/shellac films loaded with curcumin.

**Figure 5 foods-12-01577-f005:**
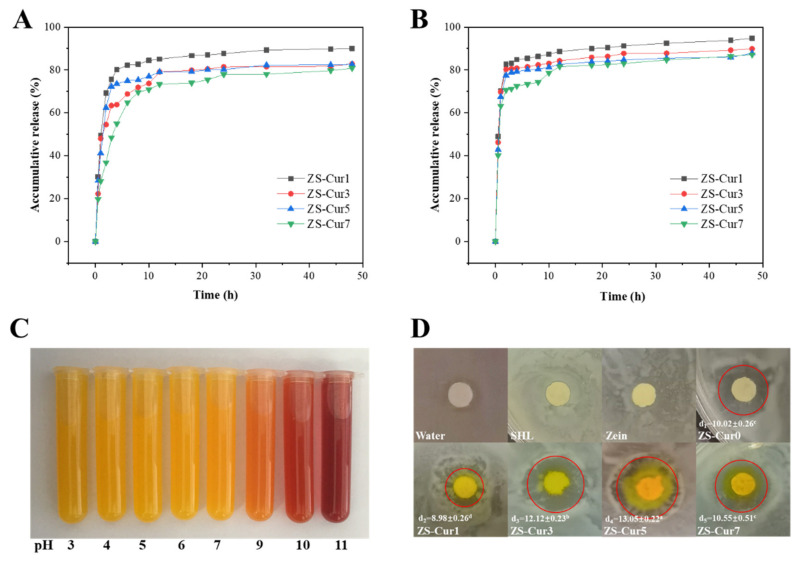
Accumulative release of curcumin from films: (**A**) semi-fatty food simulant medium, (**B**) fatty food simulant medium; (**C**) color changes of curcumin at different pH values, and (**D**) antibacterial cycle of zein/shellac films.

**Figure 6 foods-12-01577-f006:**
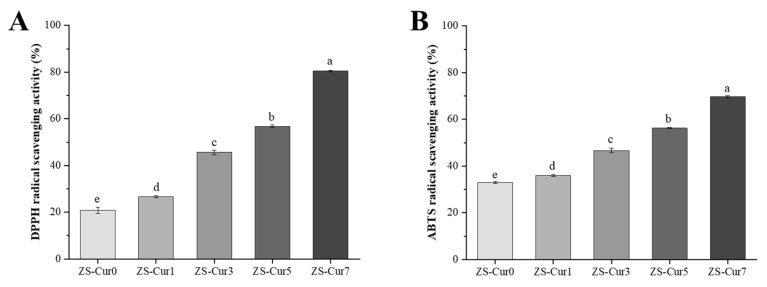
(**A**) DPPH and (**B**) ABTS radical scavenging activity of ZS–Cur films. Different superscript letters indicate significant differences (*p <* 0.05).

**Table 1 foods-12-01577-t001:** Physical properties of zein/shellac film loaded with different curcumin contents.

Sample	WVP (g∙mm∙m^−2^∙h^−1^∙kPa^−1^)	WS (%)	TS (MPa)	EB (%)
ZS–Cur0	0.405 ± 0.015 ^a^	13.266 ± 0.855 ^a^	0.725 ± 0.021 ^b^	0.642 ± 0.013 ^a^
ZS–Cur1	0.396 ± 0.007 ^ab^	13.079 ± 0.543 ^a^	0.787 ± 0.025 ^a^	0.557 ± 0.022 ^b^
ZS–Cur3	0.380 ± 0.013 ^b^	12.906 ± 0.802 ^a^	0.809 ± 0.017 ^a^	0.475 ± 0.013 ^c^
ZS–Cur5	0.388 ± 0.011 ^ab^	12.586 ± 0.651 ^a^	0.717 ± 0.036 ^b^	0.424 ± 0.016 ^d^
ZS–Cur7	0.391 ± 0.016 ^ab^	12.307 ± 0.523 ^a^	0.688 ± 0.018 ^b^	0.386 ± 0.008 ^e^

Numbers are mean ± standard deviation (n = 4). Different superscript letters within a column indicate significant differences (*p* < 0.05).

**Table 2 foods-12-01577-t002:** Optical properties of zein/shellac film loaded with different curcumin contents.

Sample	Color Parameters						Opacity
	L*	a*	b*	△E	WI	YI	
ZS–Cur0	85.60 ± 0.24 ^a^	1.27 ± 0.27 ^e^	40.54 ± 0.87 ^e^	39.68 ± 0.85 ^e^	56.96 ± 0.86 ^a^	67.66 ± 1.56 ^e^	6.20 ± 0.06 ^e^
ZS–Cur1	83.83 ± 0.63 ^b^	4.82 ± 0.65 ^d^	45.42 ± 1.80 ^d^	45.23 ± 1.63 ^d^	51.54 ± 1.56 ^b^	77.39 ± 2.66 ^d^	7.12 ± 0.16 ^d^
ZS–Cur3	78.32 ± 0.62 ^c^	12.86 ± 0.54 ^c^	50.11 ± 0.64 ^c^	52.98 ± 0.35 ^c^	43.90 ± 0.35 ^c^	91.40 ± 0.81 ^c^	7.81 ± 0.02 ^c^
ZS–Cur5	71.66 ± 0.68 ^d^	23.35 ± 0.14 ^b^	55.77 ± 0.59 ^b^	63.86 ± 0.26 ^b^	33.22 ± 0.23 ^d^	111.19 ± 0.29 ^b^	8.95 ± 0.13 ^b^
ZS–Cur7	69.77 ± 0.85 ^e^	24.82 ± 0.48 ^a^	61.78 ± 0.90 ^a^	70.20 ± 1.17 ^a^	26.88 ± 1.18 ^e^	126.54 ± 3.14 ^a^	10.02 ± 0.14 ^a^

Numbers are mean ± standard deviation (n = 4). Different superscript letters within a column indicate significant differences (*p* < 0.05). L*: lightness, a*: redness–greenness, b*: yellowness–blueness. WI: whiteness index, YI: yellowness index, and ΔE: the total color difference.

**Table 3 foods-12-01577-t003:** Water contact angle of zein/shellac films loaded with curcumin at 0 s and 60 s.

Sample	WCA (º, t = 0 s)	Images	WCA (º, t = 60 s)	Images	∆WCA (º)
ZS–Cur0	33.92 ± 1.55 ^c^	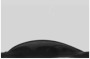	19.19 ± 0.93 ^d^	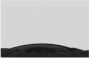	14.73
ZS–Cur1	35.46 ± 1.33 ^c^	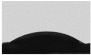	19.83 ± 1.32 ^d^	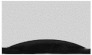	15.63
ZS–Cur3	39.33 ± 1.14 ^b^	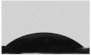	23.23 ± 0.89 ^c^	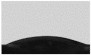	16.10
ZS–Cur5	40.4 ± 1.23 ^ab^	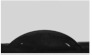	25.47 ± 1.78 ^b^	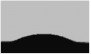	14.93
ZS–Cur7	42.1 ± 1.47 ^a^	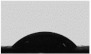	36.43 ± 1.01 ^a^	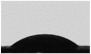	5.67

Numbers are mean ± standard deviation (n = 4). Different superscript letters within a column indicate significant differences (*p* < 0.05).

**Table 4 foods-12-01577-t004:** Weight loss of shellac, zein, and composite films at distinct stages by TGA.

	Peak 1 (°C)	Weight Loss (%)	Peak 2 (°C)	Weight Loss (%)	Peak 3 (°C)	Weight Loss (%)	Residue (%)
Shellac					438.89	65.54	8.03
ZS–Cur0	250.93	22.48	318.90	45.79	455.34	77.37	11.67
ZS–Cur1	238.63	19.71	318.81	45.78	454.24	77.10	10.68
ZS–Cur3	239.57	19.94	313.90	43.18	447.36	72.26	12.95
ZS–Cur5	226.07	16.24	310.26	43.60	453.88	75.55	13.16
ZS–Cur7	228.68	15.34	312.37	46.20	442.84	73.80	14.92
Zein			328.82	38.37			17.66

**Table 5 foods-12-01577-t005:** Color response of zein/shellac films loaded with curcumin at different pH values.

Sample	pH-Response
3	4	5	6	7	9	10	11
ZS–Cur1	Color	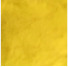	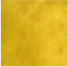	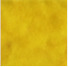	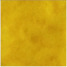	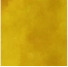	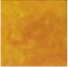	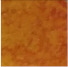	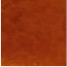
	L*	85.72 ± 1.11 ^a^	84.19 ± 0.57 ^b^	83.29 ± 0.87 ^b^	81.44 ± 0.44 ^c^	81.62 ± 0.75 ^c^	76.78 ± 0.52 ^d^	66.23 ± 1.65 ^e^	61.12 ± 0.50 ^f^
	a*	4.62 ± 1.01 ^f^	5.55 ± 0.23 ^def^	6.09 ± 1.94 ^de^	6.87 ± 0.16 ^d^	5.10 ± 0.47 ^ef^	12.02 ± 0.48 ^c^	14.60 ± 0.94 ^b^	29.65 ± 0.52 ^a^
	b*	81.18 ± 1.32 ^a^	79.08 ± 0.90 ^b^	81.04 ± 0.41 ^a^	80.07 ± 0.73 ^ab^	77.73 ± 0.83 ^c^	66.04 ± 0.92 ^d^	59.38 ± 0.39 ^e^	55.35 ± 0.53 ^f^
	△E	76.79 ± 1.18 ^a^	74.99 ± 0.90 ^b^	77.12 ± 0.59 ^a^	76.53 ± 0.68 ^a^	74.05 ± 0.80 ^b^	64.89 ± 0.89 ^d^	63.42 ± 0.97 ^e^	67.92 ± 0.65 ^c^
ZS–Cur3	Color	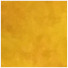	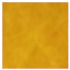	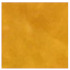	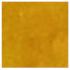	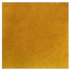	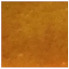	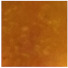	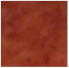
	L*	80.07 ± 0.49 ^a^	79.38 ± 0.29 ^a^	77.02 ± 0.29 ^b^	76.32 ± 0.14 ^b^	74.78 ± 0.21 ^c^	71.44 ± 0.80 ^d^	60.88 ± 0.38 ^e^	55.05 ± 1.06 ^f^
	a*	10.21 ± 0.50 ^h^	11.56 ± 0.32 ^g^	13.19 ± 0.51 ^f^	15.36 ± 0.55 ^e^	17.14 ± 0.54 ^d^	20.56 ± 0.85 ^c^	22.41 ± 0.63 ^b^	28.58 ± 0.21 ^a^
	b*	76.34 ± 0.41 ^a^	75.31 ± 0.12 ^b^	74.22 ± 0.80 ^c^	70.78 ± 0.18 ^d^	66.84 ± 0.27 ^e^	60.77 ± 0.24 ^f^	58.13 ± 0.53 ^g^	50.76 ± 0.80 ^h^
	△E	73.59 ± 0.38 ^a^	72.96 ± 0.03 ^a^	72.76 ± 0.79 ^a^	70.16 ± 0.16 ^b^	67.41 ± 0.15 ^c^	64.16 ± 0.68 ^d^	67.29 ± 0.70 ^c^	67.55 ± 0.76 ^c^
ZS–Cur5	Color	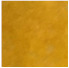	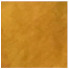	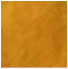	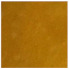	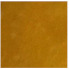	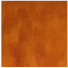	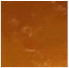	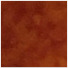
	L*	78.53 ± 0.28 ^a^	76.89 ± 0.53 ^b^	75.51 ± 0.22 ^c^	73.74 ± 0.53 ^d^	72.88 ± 0.25 ^e^	60.92 ± 0.12 ^f^	59.00 ± 0.66 ^g^	49.31 ± 1.20 ^h^
	a*	13.28 ± 0.72 ^f^	13.98 ± 0.30 ^ef^	14.55 ± 0.15 ^e^	15.36 ± 0.23 ^d^	15.89 ± 0.25 ^d^	24.53 ± 0.48 ^c^	27.98 ± 0.81 ^b^	31.87 ± 0.68 ^a^
	b*	70.06 ± 0.31 ^a^	68.68 ± 0.53 ^b^	67.31 ± 0.22 ^c^	65.05 ± 0.16 ^d^	61.93 ± 0.56 ^e^	59.53 ± 0.29 ^f^	56.39 ± 0.81 ^g^	46.82 ± 1.24 ^h^
	△E	68.46 ± 0.29 ^bc^	67.74 ± 0.37 ^cd^	66.98 ± 0.25 ^d^	65.65 ± 0.22 ^e^	63.25 ± 0.47 ^f^	69.13 ± 0.34 ^b^	69.07 ± 0.98 ^b^	70.15 ± 1.20 ^a^
ZS–Cur7	Color	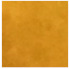	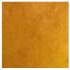	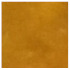	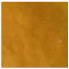	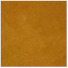	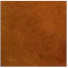	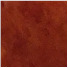	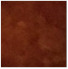
	L*	75.53 ± 0.21 ^a^	74.08 ± 0.18 ^b^	72.56 ± 0.55 ^c^	72.15 ± 0.10 ^c^	71.63 ± 0.25 ^c^	57.46 ± 0.60 ^d^	52.28 ± 0.98 ^e^	44.27 ± 1.47 ^f^
	a*	16.75 ± 0.15 ^f^	17.70 ± 0.35 ^e^	18.10 ± 0.14 ^e^	18.89 ± 0.04 ^d^	19.19 ± 0.09 ^d^	27.5 ± 0.91 ^c^	35.14 ± 0.87 ^b^	38.87 ± 0.76 ^a^
	b*	69.07 ± 0.83 ^a^	66.47 ± 0.26 ^b^	65.34 ± 0.55 ^b^	63.78 ± 0.78 ^c^	62.52 ± 0.44 ^d^	53.29 ± 1.36 ^e^	43.66 ± 1.28 ^f^	34.14 ± 0.34 ^g^
	△E	69.13 ± 0.71 ^b^	67.44 ± 0.25 ^c^	67.03 ± 0.45 ^cd^	66.00 ± 0.71 ^de^	65.17 ± 0.42 ^e^	67.45 ± 1.00 ^c^	68.02 ± 1.39 ^bc^	70.85 ± 1.36 ^a^

Numbers are mean ± standard deviation (n = 4). Different superscript letters in the same row indicate significant differences (*p* < 0.05). L*: lightness, a*: redness—greenness, b*: yellowness—blueness, ΔE: the total color difference.

## Data Availability

The data presented in this study are available in this article.

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
