# Peer review of "Development and Characterization of an Edible Zein/Shellac Composite Film Loaded with Curcumin"

_foods, 2023, doi:10.3390/foods12081577_

Round 1

Reviewer 1 Report

The authors present a well written research regarding the “Development and characterization of an edible zein/shellac composite film loaded with curcumin.”  

The global edible films and coating market size was valued at USD 2.70 billion in 2021 and is expected to expand at a compound annual growth rate of 7.7% from 2022 to 2028 (https://www.grandviewresearch.com/industry-analysis/edible-films-coating-market-report). The market for edible films and coatings is anticipated to be driven by increased consumer use of eco-friendly packaging choices. This technology can increase the quality, shelf life, safety, and usefulness of the product. Plastic is difficult to properly degrade, thus its increased usage in food packaging and transportation creates concerns.

The rising trend is justified by the significant high number of articles in the edible films domain in the Web of Science database, in the last 5 years. There has been a recent shift in focus towards substances of natural origin, mainly use of extracts from agro-industrial wastes, such as fruit and vegetable residues, as a source of bioactive compounds for the production of active edible packaging, thus contributing to the circular economy and bioeconomy.

Advantage of edible films is that they are created from components that are acceptable for human consumption and may thus be consumed alongside food products. Hydrocolloids, including polysaccharides and proteins, are the most often employed biopolymers in the creation of food materials; whether they are of animal, plant, or microbial origin, and the incorporation of different oils and fats can increase their water vapor barrier qualities. In the formulation of edible films, colorants, anti-browning agents, nutrients, spices, antibacterial and antioxidant substances can be included as active components (in present article curcumin was added to zein and shellac). When edible films convey or absorb these substances into or from the packaged food in order to increase its shelf life or enhance its condition, they might be considered as active food packaging.

The authors stated that, no studies have been reported on composite films of zein and shellac loaded with curcumin but research. Still, there are articles in the domain, that touch the subject in one form or another, among which I mention in a first reference search:

·    --Preparation, characterization and stability of curcumin-loaded zein-shellac composite colloidal particles”, by Cuixia Sun et. all. 2017,

--Food Chemistry, Musso, Y. S., Salgado, P. R., & Mauri, A. N. (2017). Smart edible films based on gelatin and curcumin. Food hydrocolloids, 66, 8-15.

--Roy, S., Priyadarshi, R., Ezati, P., & Rhim, J. W. (2022). Curcumin and its uses in active and smart food packaging applications-A comprehensive review. Food Chemistry, 375, 131885.

·   --Wang, L., Xue, J., & Zhang, Y. (2019). Preparation and characterization of curcumin loaded caseinate/zein nanocomposite film using pH-driven method. Industrial Crops and Products, 130, 71-80.

The authors realized a high number of experiments to test the newly created composite film, like water vapor perme-ability (WVP), water solubility (WS), elongation at break (EB), tensile strength (TS), water contact angle (WCA), optical characterizations by SEM, FT-IR, XRD, DSC, and TGA, pH responsiveness, antioxidant properties and inhibitory effects on E. coli. All the experiments were statistically interpreted.

Minor changes presented just need to be addressed, like:

In the abstract the author mentioned that their innovative solutions could “extend the shelf life of fresh meat”. Since meat comes with its own pH, microbial load, organic activity etc, parameters that were not mentioned in the text, I recommend replacing it with “fresh food” or similar and  leave the meat for the next practical study.

Overall great work

Author Response

Dear Editor and Reviewers,

According to your nice suggestions, we have made extensive corrections to our previous manuscript. The revised manuscript has uploaded in below.

Reviewer 2 Report

This paper investigates the development of new composite films of zein and shellac loaded with curcumin for the realization of a new edible food packaging material. The authors studied the mechanical, water/vapour barrier and optical properties of various composite films and show well supported and clear results, demonstrating the improvement of such properties upon the addition of curcumin.

Among the properties analyzed by the authors, there is the opacity of the films: the study demonstrates that the composites show good light-blocking properties and hypothesizes the capacity to protect food from oxidation. Concerning this subject, the authors should provide both the transmittance spectra of their films in the UV-Vis light range and, possibly, the materials behavior under prolonged UV irradiation time. Have the coumarin antioxidant properties any effect on the films UV resistance? The authors should refer to the following papers in order to discuss the subject: Tersili et al, Coatings 2022, 12, 417, doi:10.3390/coatings12030417; Villafiorita-Monteleone et al, Materials 2023, 16, 656, doi:10.3390/ma1602065.

Author Response

Dear Reviewer 2,

We are grateful for the suggestion. To be clearer and in accordance with the reviewer concerns, we have added the test of UV-Vis light irradiation transmittance of the films, and the detailed analysis is described in the attachment.

Reviewer 3 Report

The authors present the manuscript “Development and characterization of an edible zein/shellac composite film loaded with curcumin” as an alternative to active and functional edible films to be used in the food industry.

In my opinion, this work is very interesting and well-structured. However, there are some points that the authors should consider in order to improve the quality of the manuscript before it can be processed further in Foods.

- The manuscript is well written, but using a grammatical correction program can help to improve a few missing details.

- It is not clear how the curcumin was incorporated/dispersed into the Z/S films.  This information should be added in Section 2.3.

- The Equations should be named in the main text. Please named them before appearing in the manuscript.

- What is the meaning of the constant 142.86 in Equation 7?

- Please add the magnification and the scale used to obtain the SEM images.

- How many scans were made per sample in the FTIR analysis? Please add this information.

- In DPPH and ABTS assays are not clear quantities of the sample and reagent (DPPH or ABTS) used in the reactions. Please add this information. 

- What was the concentration (CFU/mL) of the inoculum used in the antimicrobial test? This information should be added. Moreover, add the reference on which the antimicrobial test was based.

- The antibacterial properties of the films were not named in the objective of the manuscript.  Please add this information to the phase “curcumin release ability, pH response, and antioxidant properties”

- Shellac is named along the manuscript, figures, and tables in different ways (shellac and SHL). Please uniformize it.

- It is important to name the Tables of the supplementary information in the main manuscript. 

- Add Plate count agar before (PCA) in Section 2.9

- Tables 2 and 5 are not clear. Please improve the configuration. 

Author Response

Dear Editor and Reviewers,

According to your nice suggestions, we have made extensive corrections to our previous manuscript, the detailed corrections are listed in the attachment.
